# Expression of CD44 in Leukocyte Subpopulations in Patients with Inflammatory Bowel Diseases

**DOI:** 10.3390/diagnostics12082014

**Published:** 2022-08-20

**Authors:** Ivana Franić, Nikolina Režić-Mužinić, Anita Markotić, Piero Marin Živković, Marino Vilović, Doris Rušić, Joško Božić

**Affiliations:** 1Department of Medical Laboratory Diagnostic, University Department of Health Studies, University of Split, 21000 Split, Croatia; 2Department of Medical Chemistry and Biochemistry, University of Split School of Medicine, 21000 Split, Croatia; 3Department of Gastroenterology, University Hospital of Split, 21000 Split, Croatia; 4Department of Pathophysiology, University of Split School of Medicine, 21000 Split, Croatia; 5Department of Pharmacy, University of Split School of Medicine, 21000 Split, Croatia

**Keywords:** CD14^++^CD16^−^, CD14^+^CD16^++^, and CD14^+^CD16^+^ monocytes, CD44, lymphocytes, granulocytes, Crohn’s disease, ulcerative colitis

## Abstract

CD44 expressed in monocytes and lymphocytes seems to play a crucial role in gastrointestinal inflammation, such as the one occurring in the context of inflammatory bowel diseases. Differentially methylated genes are distinctly expressed across monocyte subpopulations related to the state of Crohn’s disease. Hence, the aim of this study was to detect CD44 expression in leukocyte subpopulations in relation to the type of IBD, therapy, and disease duration. Monocyte subpopulations CD14^++^CD16^−^, CD14^++^CD16^++^, and CD14^+^CD16^+^ as well as other leukocytes were analyzed for their CD44 expression using flow cytometry in 46 patients with IBD and 48 healthy controls. Patients with Crohn’s disease treated with non-biological therapy (NBT) exhibited a lower percentage of anti-inflammatory CD14^+^CD16^++^ monocytes, whereas NBT-treated patients with ulcerative colitis had lower expression of CD44 on CD14^+^CD44^+^ lymphocytes in comparison to controls, respectively. Conversely, patients with Crohn’s disease treated with biological therapy had a higher percentage of CD44^+^ granulocytes but lower expression of CD44 on anti-inflammatory monocytes compared to controls. Median fluorescence intensity (MFI) of CD44 on CD44^+^CD14^+^ lymphocytes was higher in ulcerative colitis patients treated with biological therapy compared to NBT. The percentage of classical CD14^++^CD16^−^ monocytes was lower in the <9 years of IBD duration subgroup compared with the longer disease duration subgroup. The present study addresses the putative role of differentiation and regulation of leukocytes in tailoring IBD therapeutic regimes.

## 1. Introduction

Inflammatory bowel disease (IBD) is a group of chronic inflammatory diseases of the gastrointestinal tract, including ulcerative colitis and Crohn’s disease, with unknown etiology [1]. The pathophysiologic pathways underlying these complex clinical entities still remain elusive [2,3]. However, impaired regulation of the epithelial barrier seems to be associated with the development of IBD. It is considered that intestinal disorders result from dysfunctional epithelial, genetic structure, and abnormal innate and adaptive immune response to environmental factors [4]. Infiltration of lymphocytes and leukocytes into the lamina propria contributes to the development of intestinal inflammation, thus developing and maintaining intestinal inflammation [5].

The crucial role in gastrointestinal inflammation is attributed to CD44 expressed in monocytes and lymphocytes [6]. Flow cytometric phenotyping has identified three different monocyte subpopulations: major classical monocytes CD14^++^CD16^−^, minor non-classical CD14^+^CD16^++^, and intermediate CD14^++^CD16^+^ monocytes [7]. Their diversity can indicate pathogenesis of inflammation, whereas an increased proportion of the specific population may co-exist as a biomarker of the disease [8]. Intermediate peripheral blood monocyte population is enhanced significantly in patients with active Crohn’s disease but not ulcerative colitis [9]. After exiting from the blood and passing the endothelial barrier, leukocyte CD44 interacts with hyaluronate in the extracellular matrix [10,11,12]. Hyaluronate further participates in the organization of the extracellular matrix and is increased during inflammation.

IBD treatment is currently based on controlling the excessive immune response in the intestinal mucosa and inflammation [13]. Therapy is focused upon maintaining remission and achieving long-term better outcomes of the disease itself [14]. Non-biological therapy is still the main form of IBD treatment, especially in the inactive and mild state of the disease [15]. The strengths of NBT are safety and tolerability, maintaining remission, and low treatment costs in comparison to biological therapy treatment. On the other hand, the clinical benefits of biologic therapy are faster mucosal healing, fewer hospital interventions, and a better quality of life, while the negative aspect is the risk of serious infections and higher risk of malignancy [16].

Recently, Li Yim et al. described 12 differentially methylated promoter regions of inflammatory genes, comparing Crohn’s disease patients with active disease and those in remission [17]. A comparison of their observations with gene expression data on classical, non-classical, and intermediate monocytes indicates that seven differentially methylated genes are differentially expressed across the three monocyte subsets. Due to the relation of those genes’ expression to the state of disease, we found worthy to investigate monocyte subpopulations in differently treated IBD. The possible finding of inflammatory and anti-inflammatory monocyte profiles in relation to the therapy treatment and the state of disease could be helpful in further therapy and prognosis.

## 2. Materials and Methods

### 2.1. Study Design

The present study was organized as a cross-sectional observational study. It was conducted at the Department of Gastroenterology, University Hospital of Split and Departments of Pathophysiology and Medical Chemistry and Biochemistry, University of Split School of Medicine and the Department of Medical Laboratory Diagnostic, University Department of Health Studies, University of Split over a period from December 2017 until March 2018.

### 2.2. Subjects

The present study included 46 patients with IBD, 28 of which had Crohn’s disease and 18 ulcerative colitis. In addition, the patients were compared with a control group consisting of 48 healthy subjects. Patients with IBD were divided into two subgroups: one group treated with biological therapy and the other group treated with NBT. All 94 participants were 18 years of age or older, and each of them underwent the full study protocol except fecal calprotectin assessment, which was measured only in the patient groups. The age distribution of IBD was 40.97 ± 12.65, for ulcerative colitis 42 ± 12.3, and Crohn’s 40.32 ± 13.04 (presented as mean ± standard deviation). The patients with IBD were recruited from the outpatient clinic of the Department of Gastroenterology, University Hospital of Split. At first, 100 patients were included in the study. Among 52 patients, two were excluded due to sudden worsening of the disease, two due to inability to schedule an appointment, and finally, two due to refusal of further procedures. Subsequently, 46 patients and 48 control subjects were found to be eligible for inclusion in the study (Appendix A).

IBD diagnosis was performed in accordance with the latest guidelines of the European Crohn’s and Colitis Organization and the European Society of Gastrointestinal and Abdominal Radiology [18]. Inclusion criteria were IBD diagnosis with stable disease activity in the last 3 months and disease duration in the last 1 year. Chronic inflammatory conditions other than IBD such as chronic kidney disease, liver disease, and pulmonary disease; history of heart failure; history of cardiovascular or cerebrovascular conditions; diabetes mellitus; malignant disease; and the use of local/systemic corticosteroids in the last 3 months were exclusion criteria. Healthy blood donors from the health care centers and volunteers were included in this study. Healthy subjects were screened for the presence of the Rome IV criteria for inflammatory bowel syndrome as well as any other type of gastrointestinal symptoms.

### 2.3. Clinical Assessment and Anthropometric Measurements

All subjects underwent a detailed physical examination and anthropometric data assessment. A calibrated medical scale with an altitude meter (Seca, Birmingham, UK) was used to measure body mass and height. Body mass index (BMI) was calculated by dividing the value of body mass (kg) with the squared value of height (m^2^). Anamnestic data and tobacco consumption information were taken for all participants.

### 2.4. Assessment of Disease Severity

Disease activity in patients with IBD was assessed by two experienced gastroenterology specialists independently, using well-established clinical and endoscopic scoring systems. Disease activity was assessed using the ulcerative colitis endoscopic index of severity (UCEIS) and Mayo endoscopic score (MES) [19,20,21]. Crohn’s disease patients were assessed using the endoscopic score (SES-CD) as those with mild, moderate, or severe endoscopic form of the disease [22]. We stratified patients using only endoscopic index scores; clinical index scores were descriptively reported. We stratified different categories of ulcerative colitis and Crohn’s disease groups based on their endoscopic disease activity score.

### 2.5. Flow Cytometry

Blood samples needed for flow cytometry and biochemical analysis were collected from all participants in the morning period, after at least a 10 h fast. All samples were analyzed in a single laboratory by an experienced biochemist following the same standard procedure. The biochemist was blinded for the participant’s assignment to Crohn’s disease or ulcerative colitis group. Blood was drawn from a polyethylene catheter inserted into the antecubital vein. One hundred microliters of the whole blood was pre-treated with an Fc receptor-blocking reagent (Miltenyi Biotec GmbH, Bergisch Gladbach, Germany) to prevent nonspecific binding and was incubated for 20 min in the dark at 25 °C with 4 µL of anti-human-CD14s PerCP-Cy5.5-conjugated antibodies (eBioscience, San Diego, CA, USA), 4 µL of phycoerythrin-conjugated antibodies reactive to human CD16 (eBioscience, San Diego, CA, USA), and 10 µL of mouse antibodies reactive to human CD44 conjugated with FITC (BD Pharmingen, San Diego, CA, USA). Following the red blood cell lysis with lysis solution (Miltenyi Biotec GmbH, Bergisch Gladbach, Germany), cells were analyzed by flow cytometry (BD Accuri C6, BD Biosciences, Aalst, Belgium). Unstained cell samples were measured and processed as negative controls to set the appropriate regions. Cell acquisition was stopped at 10^6^ cells.

### 2.6. Biochemical Analysis

Fecal calprotectin concentrations were measured by turbidimetric method (Beckman Coulter AU 680), while plasma high-sensitivity C-reactive protein (hs-CRP) levels were determined using a latex turbidimetric method (Abbott Laboratories, Chicago, IL, USA). Other biochemical analyses were measured by standard laboratory methods.

### 2.7. Data Analysis

Data acquired by cytometer were analyzed using the FlowLogic Software (Inivai Technologies, Mentone Victoria, Australia). Leukocyte fluorescence is shown in the forward scatter/side scatter (FSC/SSC) dot plots. FSC parameter indicates cell diameter, and SSC indicates cell granularity. Lymphocytes population was delineated with ellipse (E1), monocyte population with (E2), and granulocytes population with (E3), respectively (Figure 1).

### 2.8. Statistical Analysis

We assessed the normality of data distribution using the Kolmogorov–Smirnov test. Data were expressed as mean ± standard deviation for continuous parametric variables, median (interquartile range) for continuous nonparametric variables, and as whole numbers and percentages for categorical variables. The Student’s *t*-test was used for the comparison of parametric continuous data and Mann–Whitney U test for the comparison of parametric continuous data. The chi-square test was used for the comparison of categorical data between groups. ANOVA Dunn’s post hoc nonparametric test was used for multiple comparisons tests.

All statistical analysis was performed using Past 3. X software (version 3.14, University of Oslo, Oslo, Norway) with the significance set at *p* < 0.05 [23].

## 3. Results

### 3.1. Basic Characteristics of the IBD and Control Group

There was no significant difference in gender, age, body weight, BMI, and smoking consumption between IBD and control group except in body height, which was lower in the IBD group (Appendix A). There was a difference in one variable, the percentage of CD14^++^CD16^−^ monocytes, which was lower in the <9 years IBD duration subgroup compared with the longer disease duration subgroup (26.82 vs. 37.09 median, *p* = 0.025) (Table 1).

Basic characteristics of two IBD, namely Chron’s disease vs. ulcerative colitis, were without statistically significant difference. There was 2-fold higher % of CD14^++^CD16^+^ monocytes in a moderate endoscopic score (*p* = 0.002) (Table 2).

Furthermore, ulcerative colitis patients had a mild-to-moderate endoscopic form of disease (UCEIS; MES) without any difference in selected variables (Table 3).

### 3.2. Comparison of Basic Anthropometric, Disease, and Laboratory Characteristics between Ulcerative Colitis and Crohn’s Disease

Selected characteristics showed no significant difference between ulcerative colitis and Crohn’s disease patients except in percentages of active smoking consumption, which was 12-fold lower (*p* = 0.006) in the ulcerative colitis group (Appendix A).

### 3.3. Basic Anthropometric and Selected Disease Characteristics between Crohn’s Disease Subgroups Regarding to Biologic Therapy

There was no significant difference in anthropometric, disease-related, and laboratory parameters between Crohn’s disease subgroups regarding to biologic therapy except erythrocyte sedimentation rate and fecal calprotectin, which were 2.4 (*p* = 0.038) and 6.1-fold lower (*p* = 0.005) in the biological therapy group, respectively (Appendix A).

### 3.4. Basic Anthropometric and Selected Disease Characteristics between Ulcerative Colitis Subgroups Regarding to Biologic Therapy

There was no significant difference in anthropometric, disease-related, and laboratory parameters between ulcerative colitis subgroups in regards to biologic therapy except white blood cells, which were lower in the NBT group (Appendix A).

### 3.5. Monocyte Subsets (CD14^++^CD16^−^, CD14^+^CD16^++^, and CD14^++^CD16^+^) Display Distinct Percentages in Differently Treated Crohn’s Disease Patients

Crohn′s disease patients treated with the non-biological treatment exhibited 2.3-fold decreased percentage of non-classical CD14^+^CD16^++^ monocytes in comparison to patients treated with biological therapy. There was no significant difference in the percentage of non-classical CD14^+^CD16^++^ monocytes and the control group (Figure 2).

There were no significant differences concerning the influence of patient treatment upon monocyte subset percentages in the ulcerative colitis patients. Median values (with interquartile ranges) for percentages of non-classical CD14^+^CD16^++^ monocytes, which were decreased in the Crohn’s non-bio group (Figure 2A), in each group were: ulcerative colitis bio, 8.545 (7.34–13.39); ulcerative colitis non-bio, 8.16 (5.39–8.9); and control 7.54 (6.15–9.27).

### 3.6. Expression of CD44 on CD14^+^CD16^++^ Monocytes in IBD

Ulcerative colitis patients treated with the biological treatment had significantly decreased median fluorescence intensity of CD44 on CD14^+^CD16^++^ monocytes in comparison to CD patients treated with biological therapy, which exhibited no significant difference in this parameter from the control group (Figure 3).

### 3.7. Expression of CD44 on CD44^+^CD14^+^ Lymphocytes in Differently Treated Ulcerative Colitis Patients

The median fluorescence intensity of CD44 on CD44^+^CD14^+^ lymphocytes was nearly 2-fold higher in ulcerative colitis patients treated with biological therapy in comparison to the NBT group. Lymphocyte percentages of the CD44^+^CD14^+^ showed no differences between the groups (Figure 4).

### 3.8. Percentage of CD44^+^ Granulocytes in Crohn’s Disease Patients

Percentages of CD44^+^ granulocytes were elevated in Crohn’s disease compared to control subjects (median = 0.6) and specifically in biologically treated patients (median = 8.3, *p* = 0.011) (Figure 5).

## 4. Discussion

Our investigation was focused on analyses of CD14, CD16, and CD44 expression in monocyte and lymphocyte subsets as well as granulocyte CD44 in patients with IBD treated with biological or NBT. Crohn′s disease patients treated with NBT showed decreased percentage of non-classical CD14^+^CD16^++^ monocytes, whereas patients with biological therapy remained at the control level. After biological treatment, decreased CD44 expression was detected in non-classical monocytes of ulcerative colitis patients, while Crohn’s disease patients’ monocytes were not affected. The percentage of classical CD14^++^CD16^−^ monocytes was lower in the <9 years of IBD duration subgroup compared with the longer disease duration subgroup. Patients treated with NBT of ulcerative colitis showed lower expression of CD44 in CD44^+^CD14^+^ lymphocytes, whereas for the patients treated with biological therapy, there was no difference in comparison to the healthy control group. The percentage of CD44^+^ granulocytes was elevated in biologically treated patients with Crohn’s disease compared to control subjects.

While we found no difference between biologically treated Crohn’s disease and healthy controls, Nazareth et al. observed a small infliximab-dependent increase in the frequency of circulating non-classical monocytes but without change compared to non-treated Crohn’s disease patients [24]. Non-classical monocytes have the highest expression of CD16 [25]. The CD16 receptor contributes to the cell activation by IgG immune complexes [26]. Consequently, non-classical monocytes are associated with adhesion, complement, and Fc gamma-mediated phagocytosis [27,28]. During IBD, the epithelial barriers of the intestines are damaged, which allows the entry of microbes into the tissue [9]. Homing of non-classical monocytes via α4β7 integrin to the gut mediates macrophage-dependent intestinal wound healing [29]. These macrophages are upregulated in the proliferative phase of wound healing but decreased following vedolizumab, an anti-α4β7 integrin antibody routinely used in IBD treatment [30,31]. According to Nazareth et al. [18], infliximab corrects the defective TNF production of Crohn’s disease macrophages, which seemed to depend upon the enrichment of CD16^+^ circulating monocytes (including non-classical CD14^+^CD16^++^ subset).

Therapies interfering with gut homing have an impact on monocyte subset homing. Classical monocytes preferentially home to inflamed sites in contrast to the non-classical monocytes that home to non-inflamed tissues [32]. In peripheral tissues, monocytes may develop into macrophages and monocyte-derived dendritic cells [33]. The presence of cytokines and microbial compounds can give rise to various types of macrophages: proinflammatory, which secrete cytokines such as TNF-α, IL-12, or IL-23, and anti-inflammatory, which produce cytokines such as IL-10 or TGF-β [34]. The general perception is that non-classical monocytes are biased towards wound-healing macrophages [29]. Decreased percentage of anti-inflammatory non-classical CD14^+^CD16^++^ monocytes in non-biologically treated Crohn’s disease and their unchanged level after biological therapy indicate the higher potential of intestinal wound healing in biologically treated patients. In non-biologically treated ulcerative colitis, percentage of non-classical monocytes remained as in the control group. Different production of non-classical CD14^+^CD16^++^ monocytes between two IBD could be the consequence of different interplay between leukocytes and tissue cells. Ulcerative colitis and Crohn’s disease differ concerning the IL-18 involvement. IL-18 induces only Crohn’s disease, but not in all patients [35], enhancing NK cell cytotoxicity and stimulating Th1 responses [36,37].

Biological treatment decreased CD44 expression on non-classical monocytes of ulcerative colitis patients, while monocytes in patients with Crohn’s disease were not affected. Total monocyte CD44 isoforms are more precisely designated as CD44s, containing variant exons (CD44v and CD44v7) in human-activated monocytes and different T-cell subpopulations [38]. In addition to hyaluronan [10,11,12], partner ligand molecules of CD44 receptor include laminin, hepatocyte growth factor, vascular endothelial growth factor, and osteopontin [38,39]. Inflammation is initiated by elevated IL-6 secretion from monocytes induced after interaction between osteopontin and CD44v7 [36]. In CD44v7 knock-out mice, IL-6 levels are 30 times lower compared to wild-type mice, leading to reduced colonic inflammation. Expression of CD44 gene was higher in patient biopsies taken from inflamed than from non-inflamed regions [39]. CD44 binds to hyaluronan, laminin, and osteopontin within the extracellular matrix after monocyte extravasation [38,39]. Prior to the extravasation, monocyte ligand α4β7 has to bind to the sMAdCAM-1 receptor on endothelial cell [40]. A phase II study of fully human antibody towards MAdCAM-1 showed induction of remission and mucosal healing after 12 weeks in patients with ulcerative colitis but not with Crohn’s disease, probably due to an inflammatory change that extends through the entire bowel wall [40]. We can assume the same for the unaffected CD44 expression on non-classical monocytes in Crohn’s disease patients in this study.

Classical CD14^++^CD16^−^ monocytes are inflammatory [41]. Elevated secretion or dysregulation of IL-6 and its signaling pathway may play a major role in the pathogenesis of IBD [39,40,42]. With longer disease duration, inflamed colonic mucosa exhibits increasing chromosomal instability and hypermethylation, marking the colon at risk for further carcinogenesis and indicating the important role of inflammation in this setting [43,44]. Increased risk for colorectal cancer is related to the effect of colon mucosal inflammation, especially after 8 to 10 years [45]. In the present study, percentage of CD14^++^CD16^−^ monocytes was lower in the <9 years of IBD duration subgroup compared with the longer disease duration subgroup, indicating better control of the disease in the first period.

In parallel to the monocyte analyses, CD14, CD16, and CD44 surface molecules were also analyzed in lymphocytes [46]. Lymphocytes exhibit low levels of CD14 staining. Among 21 different lymphocyte subpopulations, Kalina et al. detected CD14 only at naive B tonsil cells [47]. Naive B cells continuously recirculate throughout the body [48]. Upon extravasation from the peripheral circulation through endothelial vesicles found in the lymphoepithelium into the secondary lymphoid tissue, such as the tonsils, B cells remain there for a few days and then re-enter the circulation unless they encounter cognate antigen [28]. In this study, we detected higher CD44 expression in minor CD44^+^CD14^+^ lymphocyte subpopulation in biologically treated ulcerative colitis compared to NBT. NBT resulted in lower CD44 expression compared to the control group, while the biologically treated and control group had similar CD44 expression. In ulcerative colitis, blood lymphocytes include circulating naive and memory T and B lymphocytes as well as primed effector cells that are *en route* to the inflamed gut mucosa [49]. Naive cells constitute more than half of the blood B cells in healthy adults. Patients with ulcerative colitis and those with Crohn’s disease show increased percentages of CD23^+^ B cells, which are recognized as being naive B cells [50]. However, in ulcerative colitis, gut inflammation and activation of T cells appear to be a hallmark of the disease. In Crohn’s disease, arrested B-cell activation beyond the naive stage seems to be a characteristic sign [44]. Colonic mucosa samples from patients with ulcerative colitis lack naive B cells that are present in samples from patients with active Crohn’s disease [50]. We can speculate that lower CD44 expression in CD44^+^CD14^+^ lymphocyte in blood after NBT, found in ulcerative colitis but not in Crohn’s disease, can be due to the higher naive B-cell count in ulcerative colitis. Excessive production of IL-1, IL-6, IL12/23, and TNFα mediate distinct abnormal T-cell responses resulting in tissue fibrosis [51].

The expression of CD44 per one granulocyte was unaffected in both IBD groups of our study, but the percentage of CD44^+^ granulocytes was elevated in Crohn’s disease compared to control and specifically in biologically treated patients with Crohn’s disease. Lampinen et al. described a significantly larger percentage of CD44^high^ eosinophils in patients with active Crohn’s disease and ulcerative colitis compared with control subjects [52]. CD44^high^ eosinophils from collagenous colitis intestinal biopsy samples have higher CD66b expression in activated state. Budesonide treatment restores the normal activation of eosinophils [53]. The role of neutrophils in the gut differs between Crohn’s disease and ulcerative colitis [29]. In ulcerative colitis, unrestricted neutrophil activation cause tissue damage, whereas in Crohn’s disease, defective neutrophils are not able to limit invasion by microorganisms, leading to uncontrolled inflammatory reaction [54,55,56].

Biological therapy lowered erythrocyte sedimentation rate (ESR) and fecal calprotectin in ulcerative colitis in comparison to control, while these parameters were unchanged in patients with Crohn’s disease. Furthermore, elevated fecal calprotectin was positively associated with inflammatory biomarkers, namely ESR and CRP [51]. Higher fecal calprotectin and the presence of mucus in diarrhea are known as significant predictors of the development of Crohn’s disease and disease activity [57,58,59]. In the multivariate analyses, only monocyte count and fecal calprotectin were associated with relapse [60]. NBT lowered white blood cells in patients with ulcerative colitis in our study, rendering them more vulnerable to infections.

Biological therapy represents a significant improvement in the treatment of IBD, as it can inhibit the inflammatory cascade in chronic inflammatory disease and thus change the course and stop the progression of the disease. However, the possible side effects are numerous and can affect almost any organ system, some of which can be very serious, such as the development of serious infections, pulmonary embolism of systemic and respiratory hypersensitivity, gastrointestinal fistulas, and malignant and lymphoproliferative diseases.

Understanding the fundamental differentiation and regulation of leukocyte cells will dictate future therapeutic effects, reducing their effect when they are harmful and amplifying their effect when they are useful. This study has several limitations. The design of the present study was cross-sectional, which impedes the establishment of a true causal relationship. Second, this study had a relatively small number of participants. The IBD group from our study mostly was heterogenous in disease activity, which additionally decreases the sample size by stratification and impedes the generalization of the findings to the entire IBD population. IBD is a repeating disease, and the pattern of excreted markers is not constant, so one measurement is not enough to determine a causal relationship. In the preparation of the study, we used only the parameter hs-CRP; other inflammatory parameters that play an important role in the results in the preparation of study were not measured.

## 5. Conclusions

We describe distinct anti-inflammatory and inflammatory CD44^+^ monocyte profiles in relation to the types of IBD and their therapies. CD44^+^CD14^+^ lymphocytes and CD44^+^ granulocytes were affected as well. Due to CD44 interactions with extracellular matrix, its optimal expression can be crucial in intestinal wound healing. Nevertheless, future experiments are needed to distinguish between circulating naive and memory T and B lymphocytes, particularly in ulcerative colitis. Independently of the therapy and type of IBD, the percentage of inflammatory monocytes was dependent upon disease duration. This finding emphasizes the importance of inflammatory monocyte monitoring.

## Figures and Tables

**Figure 1 diagnostics-12-02014-f001:**
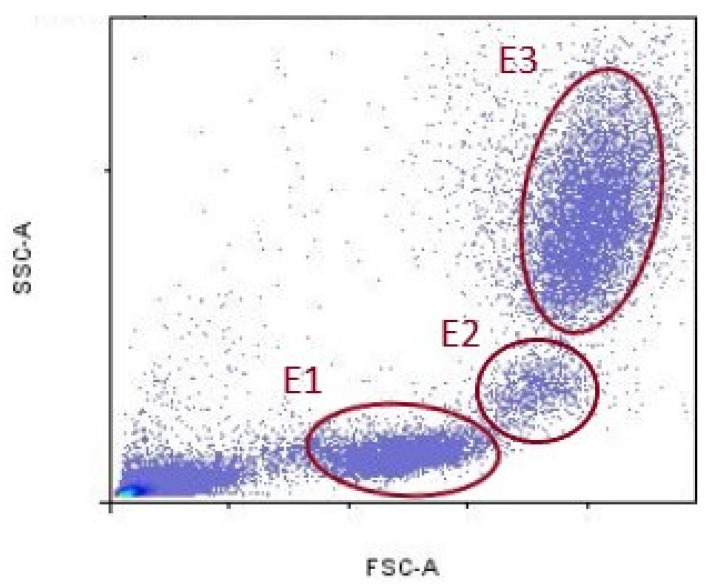
Representative gates for lymphocytes (E1), monocytes (E2), and granulocytes (E3).

**Figure 2 diagnostics-12-02014-f002:**
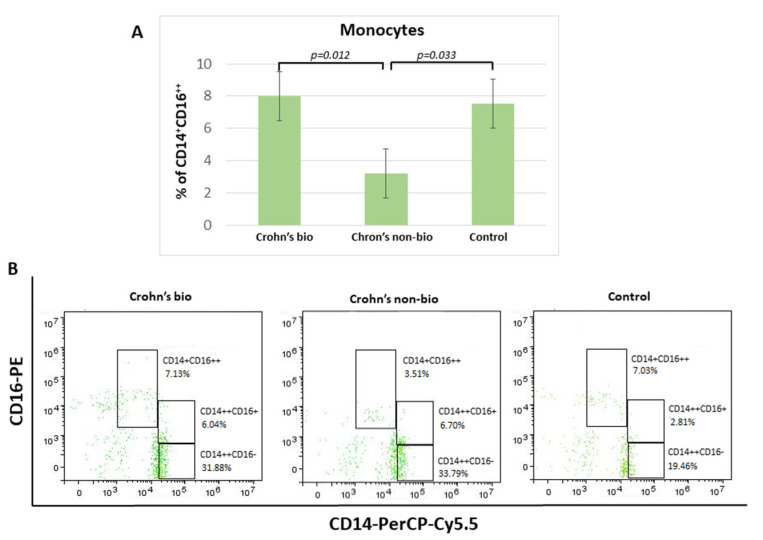
Percentage of monocyte subpopulations (CD14^++^CD16^−^, CD14^+^CD16^++^, and CD14^++^CD16^+^) in patients with IBD. Statistical histograms (**A**) and representative dot plots (**B**) of patients treated with biological and non-biological therapy as well as of control subjects. *p* < 0.05.

**Figure 3 diagnostics-12-02014-f003:**
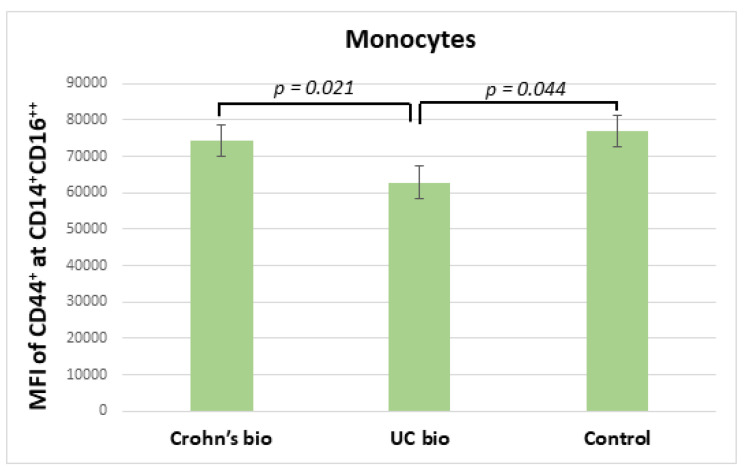
MFI of CD44 on CD14^+^CD16^++^ monocytes in IBD biological therapy treated patients and control subjects. MFI, median fluorescence intensity; UC, ulcerative colitis.

**Figure 4 diagnostics-12-02014-f004:**
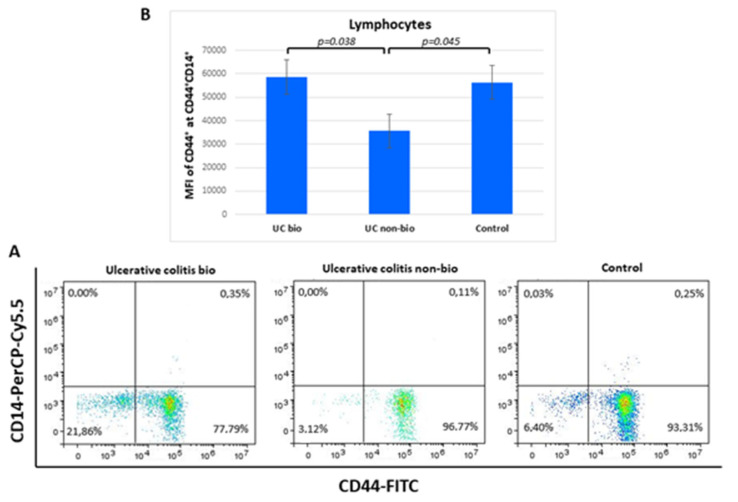
Percentage of CD44^+^CD14^+^ lymphocytes (**A**) and their CD44 median fluorescence intensity (**B**) in ulcerative colitis patients. MFI, *median fluorescence intensity*.

**Figure 5 diagnostics-12-02014-f005:**
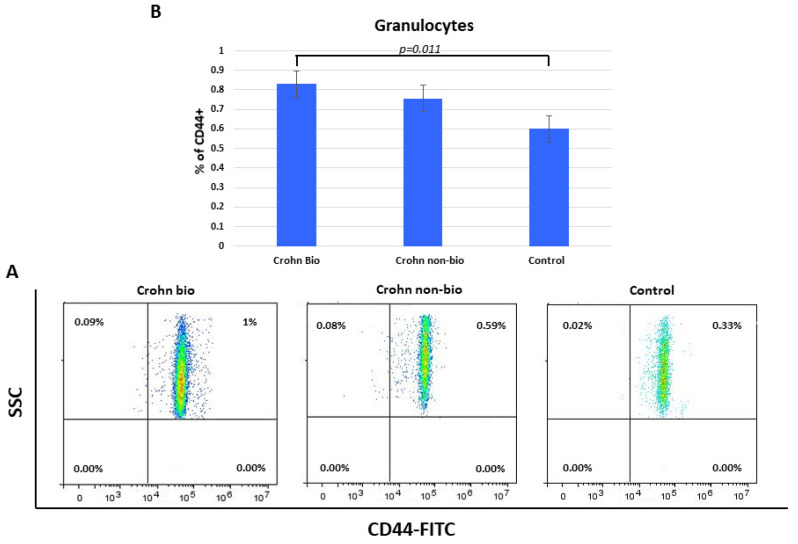
Percentage of CD44+ granulocytes in patients with Crohn’s disease. Representative dot plots (**A**) and statistical histogram (**B**) of patients treated with biological and non-biological therapy as well as of control subjects.

**Table 1 diagnostics-12-02014-t001:** Comparison of selected parameters between IBD subgroups according to median of disease duration.

Parameters						*p **
	Disease Duration					
	<9 Years (*n* = 25)		>9 Years (*n* = 20)			
Lymphocytes MFI of CD44 at CD44^+^CD14^+^	51,894 (44,287–67,815)		58,532 (35,226–71,311)			0.578
Lymphocytes MFI of CD44 at CD44^+^CD16^+^	21,891 (15,000–34,714)		26,822 (12,568–39,607)			0.195
Monocytes % of CD14^+^CD16^++^	7.64 (4.28–9.93)		7.45 (5.81–10.33)			0.911
Monocytes % of CD14^++^CD16^+^	6.9 (5.85–13.28)		10.52 (7.56–16.27)			0.149
Monocytes % of CD14^++^CD16^−^	26.82 (20.49–33.39)		37.09 (25.56–45.84)			0.025
MFI of CD44 at CD14^+^CD16^++^	71,165 (64,885–84,584)		73,558 (61,522–89,867)			0.679
MFI of CD44 at CD14^++^CD16^+^	117,277 (85,335–128,315)		128,843 (96,248–140,698)			0.122

Date are presented as median (interquartile range). LYM, lymphocyte; MFI, median fluorescence intensity. * Mann–Whitney U test.

**Table 2 diagnostics-12-02014-t002:** Comparison of selected parameters between different disease activity categories in Crohn’s disease patients.

	Endoscopic Disease Activity (SES-CD)										
Parameters	Mild			Moderate			Severe			ANOVA	
	(*n* = 9)			(*n* = 9)			(*n* = 9)			F	*p* *
**Lymphocytes**											
MFI of CD44 at CD44^+^CD14^+^	89,316 (76,000–104,839)			81,400 (58,117–95,505)			70,499 (51,503–114,550)			0.55	0.55
**Monocytes**											
CD14^+^CD16^++^ %	5.99 (1.9–7.25)			7.98 (6.48–11.63)			5.95 (2.76–11.3)			1.56	0.23
CD14^++^CD16^+^ %	7.69 (6.45–9.28)			16.04 (9.65–19.7)			6.43 (4.67–9.7)			8.03	0.002
CD14^++^CD16^−^ %	33.8 (23.95–46.93)			27.92 (25.7–36.9)			35.4 (24.75–39.36)			0.36	0.7
MFI of CD44^+^ at CD14^++^CD16^−^	90,675 (71,777–92,797)			84,560 (69,068–94,359)			85,768 (73,743–97,878)			0.44	0.644
MFI of CD44^+^ at CD14^+^CD16^++^	81,708 (74,904–92,364)			69,864 (56,979–71,249)			90,720 (71,249–103,679)			0.77	0.471

Date were presented as median (interquartile range). SES-CD, simple endoscopic score for Crohn’s disease; MFI, median fluorescence intensity; CD, cluster of differentiation. * ANOVA. Dunn’s post hoc test.

**Table 3 diagnostics-12-02014-t003:** Comparison of selected parameters between different disease activity categories in ulcerative colitis patients.

	Endoscopic Disease Activity (UCEIS; MES)						
Parameters	Mild			Moderate			*p* *
	(*n* = 5)			(*n* = 11)			
**Lymphocytes**							
MFI of CD44 at CD44^+^CD14^+^	25,028 (13,845–104,526)			69,528 (66,467–94,765)			0.364
**Monocytes**							
CD14^+^CD16^++^ %	8.9 (6.82–10.14)			8.77 (6.37–14.24)			0.82
CD14^++^CD16^+^ %	7.71 (4.53–16.34)			7.66 (5.38–12.9)			0.89
CD14^++^CD16^−^ %	25.53 (22.52–43.44)			25.26 (15.26–32.93)			0.42
MFI of CD44^+^ at CD14^++^CD16^−^	97,363 (75,098–98,412)			76,118 (62,665–86,808)			0.058
MFI of CD44^+^ at CD14^+^CD16^++^	91,715 (72,015–113,863)			68,445 (26,607–86,346)			0.302

Date were presented as median (interquartile range). UCEIS, ulcerative colitis endoscopic index of severity; MES, Mayo endoscopic score; MFI, median fluorescence intensity; CD, cluster of differentiation. * Mann–Whitney U test.

## Data Availability

Further information regarding the resources and data availability should be directed to the corresponding author.

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
