# Peer review of "Expression of CD44 in Leukocyte Subpopulations in Patients with Inflammatory Bowel Diseases"

_diagnostics, 2022, doi:10.3390/diagnostics12082014_

Round 1

Reviewer 1 Report

The paper entitle "Expression of CD44 in leukocyte subpopulations in patients with inflammatory bowel diseases" addresses the putative role of differentiation and regulation of leukocytes in tailoring IBD therapeutic regimes. I think that it could be accept in present form; the work is well structured, the research design is appropriate, the methods and the results are clearly described and the discussion and conclusions are well supported by the results.

Author Response

Response to Reviewer 1 Comments

Point 1: The paper entitle "Expression of CD44 in leukocyte subpopulations in patients with inflammatory bowel diseases" addresses the putative role of differentiation and regulation of leukocytes in tailoring IBD therapeutic regimes. I think that it could be accept in present form; the work is well structured, the research design is appropriate, the methods and the results are clearly described and the discussion and conclusions are well supported by the results.

Response 1: We thank the honorable reviewer for the time spent reviewing our paper. Also, we are grateful for the positive comments.

Reviewer 2 Report

The manuscript entitled "Expression of CD44 in leukocyte subpopulations in patients with inflammatory bowel diseases" focuses on detection of CD44 expression in leukocyte subpopulations with regard to to the type of IBD, therapy, and disease duration. The authors show how the monocyte population is differentially regulated in Crohn's disease and Ulcerative colitis. In addition, how NBT differentially affects the leukocyte population in CD and UC. The study holds high importance as it is carried out in human patient samples and also has a good sample size.

Author Response

Response to Reviewer 2

Point 1: The manuscript entitled "Expression of CD44 in leukocyte subpopulations in patients with inflammatory bowel diseases" focuses on detection of CD44 expression in leukocyte subpopulations with regard to to the type of IBD, therapy, and disease duration. The authors show how the monocyte population is differentially regulated in Crohn's disease and Ulcerative colitis. In addition, how NBT differentially affects the leukocyte population in CD and UC. The study holds high importance as it is carried out in human patient samples and also has a good sample size. 

Response 1: We thank the honorable reviewer 2 for the time spent reviewing our paper. Also, we are grateful for the positive comments.
